# 3-D cephalometry of the the orbit regarding endocrine orbitopathy, exophthalmos, and sex

**Konstantin Volker Hierl**[1]*, **Matthias Krause**[2], **Daniel Kruber**[3], **Ina Sterker**[1]

**1** Department of Ophthalmology, Leipzig University, Leipzig, Germany, **2** Department of Oral & Maxillofacial Plastic Surgery, Leipzig University, Leipzig, Germany, **3** Department of Informatics and Media, Leipzig University of Applied Sciences, Leipzig, Germany

* konstantin.hierl@gmx.de

## Abstract

### Purpose

This study aimed at evaluating the orbital anatomy of patients concerning the relevance of orbital anatomy in the etiology of EO (endocrine orbitopathy) and exophthalmos utilizing a novel approach regarding three-dimensional measurements. Furthermore, sexual dimorphism in orbital anatomy was analyzed.

### Methods

Orbital anatomy of 123 Caucasian patients (52 with EO, 71 without EO) was examined using computed tomographic data and FAT software for 3-D cephalometry. Using 56 anatomical landmarks, 20 angles and 155 distances were measured. MEDAS software was used for performing connected and unconnected t-tests and Spearman´s rank correlation test to evaluate interrelations and differences.

### Results

Orbital anatomy was highly symmetrical with a mean side difference of 0.3 mm for distances and 0.6˚ for angles. There was a small albeit statistically significant difference in 13 out of 155 distances in women and 1 in men concerning patients with and without EO. Two out of 12 angles showed a statistically significant difference between female patients with and without EO. Regarding sex, statistically significant differences occurred in 39 distances, orbit volume, orbit surface, and 2 angles. On average, measurements were larger in men. Concerning globe position within the orbit, larger distances to the orbital apex correlated with larger orbital dimensions whereas the sagittal position of the orbital rim defined Hertel values.

### Conclusion

In this study, little difference in orbital anatomy between patients with and without EO was found. Concerning sex, orbital anatomy differed significantly with men presenting larger orbital dimensions. Regarding clinically measured exophthalmos, orbital aperture anatomy

**Data Availability Statement:** All relevant data are within the paper and its Supporting Information files.

**Funding:** The authors received no specific funding for this work. The development of the 3-D analysis

software (FAT) is partially funded by German Federal Ministry of Economics and Technology ZIM KF2036708SS0. The funders had no role in study design, data collection and analysis, decision to publish, or preparation of the manuscript.

**Competing interests:** The authors have declared that no competing interests exist.

is an important factor which has to be considered in distinguishing between true exophthalmos with a larger distance between globe and orbital apex and pseudoexophthalmos were only the orbital rim is retruded. Thus, orbital anatomy may influence therapy regarding timing and surgical procedures as it affects exophthalmos.

## Introduction

Endocrine orbitopathy (EO) is an inflammatory autoimmune disease affecting the orbit occurring in 16/100,000 women and 2.9/100,000 men per year with an onset between 30 and 60 years [1]. EO is typically associated with Graves' disease and one of its most relevant extrathyroidal manifestations but may also occur in association with other diseases of the thyroid [2].

Characteristic symptoms in EO include exophthalmos, upper eyelid retraction, chemosis, conjunctival injection, and diplopia. Loss of vision due to optic neuropathy is a feared complication [3].

The most important aspects in the management of EO are the restoration and maintainance of euthyreoidism as well as immunosuppressive therapy [4]. Rehabilitative surgery is an option in stable and inactive EO as well as in vision-threatening EO [4].

Many surgical methods for decompression (i.e. resection of one to four orbital walls with or without orbital fat removal) have been established since its first description by Dollinger in 1911 [5–7]. All surgical therapies necessitate the knowledge of orbital anatomy and the anatomy in EO for surgery planning and the evaluation of surgical success.

Utilizing measurements in CT-scans, Baujat et al. and Rajabi et al. [8,9] did not find significant anatomical differences in EO patients besides slight differences in the lateral orbital angle (angle between the midsagittal plane and the lateral orbital wall) or interorbital distance. These studies, however, were limited by several factors like the use of axial CT slices for 2-D measurements instead of 3-D cephalometry [8,9]. Baujat et al. compared two patient groups with exophthalmos (EO and non-EO) [8] whereas Rajabi et al. evaluated a small groups of non-EO patients [9]. The number of observed parameters was limited in both evaluations. Moreover, the influence of sex was not evaluated.

Thus, the purpose of our study was to evaluate orbital anatomy in patients with and without EO investigating the relevance of orbital anatomy in the etiology of EO and exophthalmos–one of its major clinical features–using a new approach of three-dimensional cephalometric measurement. As EO occurs more often in women, we also studied anatomical differences between men and women. While such differences have been described before [10,11], we included more anatomical parameters than previous observations and utilized the possibilities of 3-D cephalometry. Beyond that, we evaluated the influence of anatomical traits on the extent of exophthalmos in EO and non-EO patients as these anatomical factors could contribute to patients seeking treatment. Regarding surgery in EO patients, these potential traits could be useful in deciding on appropriate surgical procedures. Finally, we evaluated the symmetry of orbital anatomy besides overall orbital shape as methods based on mirroring and using the unaffected side as a model [12] are important in surgical reconstruction concerning orbital trauma and disease.

## Materials and methods

The study was based on cephalometric CT scan analysis. The CT scans without anatomical discernable pathology (reference group) had been acquired in search for foci in the head &

neck region due to general illnesses, in search for anatomical causes of neurologic disorders, and in preparation of oral surgery or oral and maxillofacial surgery not associated with anatomical alterations relevant to this investigation. Besides the absence of other pathologies of the orbit, the CTs had to have a slice thickness of not more than 1.5 mm with contiguous slicing to be included in our study.

To measure the orbital volume and surface, we imported the CT data into iPlan Brainlab (Brainlab AG, Feldkirchen, Germany) software and used automatic segmentation with subsequent manual adjustments [13]. Subsequently, we imported the resulting STL (Standard Triangulation Language) files into FAT (Facial Analysis Tool) software to measure surface and volume. All 3-D measurements were made in FAT software [14,15]. We developed a cephalometric analysis using 56 landmarks (5 unilateral landmarks, 21 bilateral landmarks, 3 unilateral constructed landmarks, 3 bilateral; Table 1, Fig 1), 14 planes, 20 angles and measured 155 distances (70 bilateral, 15 unilateral) (Table 2, Fig 2). Landmarks were placed either directly on the 3-D object or by using crosshairs in landmarks not associated with bone. Crosshairs could be utilized in the 3-D reconstruction or on three multiplanar reformatted (MPR) image planes which could be overlaid with the 3-D object (which then could be faded out). Thus, the center of the optic canal was marked on the 3-D display, whereas *Sella* point was defined on the overlaid MPRs in the 3-D window, while the location of the landmarks was additionally shown on each plane in a separate window. The cephalometric analysis was created by using a construction matrix using landmarks to define lines, planes, distances, and angles [14]. The great advantage of this approach is that in having landmark data sets and an analysis matrix, modifications within the landmark data or analysis can be performed without repeating the whole data acquisition. The analysis results were exported into an excel spreadsheet and then imported into the statistical analysis software.

We evaluated computed tomographic data of 123 adult Caucasian patients. Out of those patients, 52 (42.3%) had known EO and were scheduled for decompression surgery. 71 patients (57.7%) had no known pathology of the orbit (e.g. trauma, tumors, deformities) and constituted our reference group.

Statistical analysis was then performed with Winmedas (Fa. Christian Grund, Margetshoechheim, Germany) software. We assessed the anatomical data of patients with and without known EO as well as male and female patients in unconnected t-tests and the symmetry between left and right orbit in connected t-tests. Correlations were measured using Spearman's rank correlation test.

Results with $p \leq 0.05$ were regarded as statistically significant.

This retrospective study was approved by the Leipzig University ethics commission (No. 285-14-25082014) and followed the Declaration of Helsinki on medical protocols and ethics. As pseudonymization of the CT data was performed, the requirement of informed consent was waived by the ethics committee.

## Results

Average age of the total patient group was 44.0 ±15.7 years (range 18–84 years). Thirty-five patients with known EO (67.3%) were female and 17 (32.7%) male. Their mean age was 49.1 ±10.4 years (range 28–76 years). Our reference group of 71 patients consisted of 46 (64.8%) male and 25 (35.2%) female patients with a mean age of 40.2 ± 17.8 years (range 18–84 years).

Globally, the orbital anatomy measures proved to be highly symmetrical with an average side difference in distances of 0.3 mm and no mean difference in any distance greater than 0.6 mm. The average side difference in angles was 0.6˚ with no mean side difference being greater than 1.7˚. This observation was valid in all groups.

**Table 1. 3-D landmarks placed in FAT and their respective definitions, constructed planes and angles.**

| Anatomical landmarks | | |
|---|---|---|
| Apex orbitae (bilateral) | AO | central point of optic canal at the border of the orbit |
| Apex orbitae fissure (bilateral) | Aof | most medial caudal point of superior orbital fissure |
| A-Point | A | most retral point of the curvature of the upper alveolus (in MSP plane) |
| Basion | Ba | most anterior point of foramen magnum in midsagittal plane |
| Dakryon (bilateral) | Da | contact point of maxilla,frontal bone, and lacrimal bone |
| | | (contact anterior lacrimal crest and maxillofrontal suture) |
| Frontomalar suture (bilateral) | Fms | medial-anterior point of frontozygomatic suture |
| Midpoint lateral orbital rim (bilat.) | Mlo | midpoint of lateral orbital rim |
| Infraorbitale (bilateral) | In | most superior point of infraorbital foramen |
| Infraorbital midpoint (bilateral) | Inf | midpoint of intraorbital rim |
| Infraorbital rim lateral (bilateral) | Il | most lateral point of infraorbital rim |
| Infraorbital rim medial (bilateral) | Im | most medial point of infraorbital rim |
| Orbitale (bilateral) | Or | most caudal point on infraorbital rim |
| Nasion (bone) | N | midpoint of fronto-nasal suture |
| Porion (bilateral) | Po | most superior point of the external auditory meatus |
| Sella | S | center of sella turcica |
| Anterior sphenoid trigone (bilat.) | Ast | anterior border of the sphenoid trigone (level of the midpoint of the lateral orbital wall) |
| Posterior sphenoid trigone (bilat.) | Pst | posterior border of the sphenoid trigone (level of the midpoint of the lateral orbital wall) |
| Supraorbitale (soft tissue) bilat.) | So | point overlying bony Suparorbitale |
| Supraorbital foramen (bilateral) | SupF | central-caudal point of supraorbital foramen |
| Supraorbital midpoint (bilateral) | SupM | midpoint of supraorbital rim |
| Globe anterior (bilateral) | Ga | most anterior point of globe diameter |
| Globe posterior (bilateral) | Gp | most posterior point of globe diameter |
| Orbitale soft tissue (bilateral) | Or | soft tissue point overlying bony orbitale |
| Exokanthion (bilateral) | Exo | most lateral point of palpebral fissure |
| Endokanthion (bilateral) | Endo | most medial point of palpebral fissure |
| Nasion soft tissue | N | point overlying bony Nasion |
| **Constructed landmarks** | | |
| Midaperture (bilateral) | mA | midpoint of orbital aperture; lines connecting (Inf-SupM) and (Mlo-Da) |
| Midpoint globe (bilateral) | mG | midpoint of globe |
| Midporion | mPo | midpoint between left-right Porion |
| Midpoint Mlo | mMlo | midpoint left-right midpoint lateral orbital rim |
| Midapex | mApex | midpoint Aof bilateral |
| Midpoint lid aperture (bilateral) | mL | midpont-exo-endo |
| **Planes** | | |
| Frankfurt Horizontal | FH | Porion right- Porion left—Infraorbitale left |
| Midsagittal plane | MSP | perpendicular to FH through N and mPo |
| Coronar plane | Cor | perpendicular to FH and MSP, through right Porion |
| parallel planes to Cor | Corpara-... | through landmarks Da, SupM, Ga, Mlo, AO, Aof, Ma, Inf, S, Ba, Fms |
| parallel planes to FH | FHpara... | through landmarks Mlo, ... |
| **Angles** | | |
| SNA° | | internal angle between S, N, and A landmarks |
| BaSN° | | internal angle between Ba, S, and N landmarks |
| latorbangle ° | | internal angle between left-right Fms and S |
| medorbangle° | | internal angle between left-right Da and S |
| latorb-MSP° (bilateral) | | internal angle between Mlo-S and MSP |
| latorb-MSP2° (bilateral) | | internal angle between Mlo-AO and MSP |

*(Continued)*

**Table 1.** (Continued)

| | |
|---|---|
| Horizontal orbit angle˚ (bilateral) | internal angle between lateral and medial orbital walls (Mlo-AO and Da-AO) |
| Vertical orbital angle˚ (bilateral) | internal angle between SupM-AoF and Inf-AoF |
| Infraorb slope˚ (bilateral) | internal angle Inf-AoF and FH |
| lateral orbital slope˚ (bilateral) | internal angle between line left-right Mlo and Da (left/right)-Mlo (left/right) |
| Orbithorizontal-Cor˚ (bilateral) | internal angle between Da-Mlo and Cor |
| Globe axis˚ (bilateral) | internal angle Ga-Gp and MSP |

The comparison of the orbital anatomy in patients with and without EO showed statistically significant differences in 13 out of 155 distances (Table 3) in women and in one distance in men. Women with EO had a larger orbital volume and concomitantly orbital surface area which was related to a larger orbital height, width, and length. The *medorbangle* (measured between left and right Dakryon and Sella) was smaller in women with EO which originated from a smaller anterior interorbital width (distance left–right *Dakryon*). Women with EO presented a larger *horizontal orbit angle* (the angle between medial and lateral orbital walls). Furthermore, *Orbitale* (most caudal landmark on the infraorbital rim) was positioned more anterior in female EO patients. Albeit statistically significant different, those differences were relatively small.

While these parameters showed no statistically significant differences in men, the position of the anterior border of the *sphenoid trigone* was more anterior in men with EO (the only specifically male difference), while the length of the sphenoid trigone was increased in women with EO. Without discerning sex, only *orbital height*, *sphenoid trigone length*, and *horizontal orbital angle* were increased in EO, while the *posterior interorbital width* (measured at *Apex orbitae*) was decreased.

Concerning sex, distinct differences were found. Table 4 shows these seen in both EO and non-EO patients regarding hard tissue measurements. Thus, values for all patients are stated for bony measurements. As soft tissue values are influenced by EO only the results of the non-EO group are given. Men showed a significantly larger orbital volume and surface area which was related to a larger orbital width, depth, and longer orbital walls with a more anterior positioned lateral orbital rim midpoint. Men presented a more anterior supraorbital rim (in

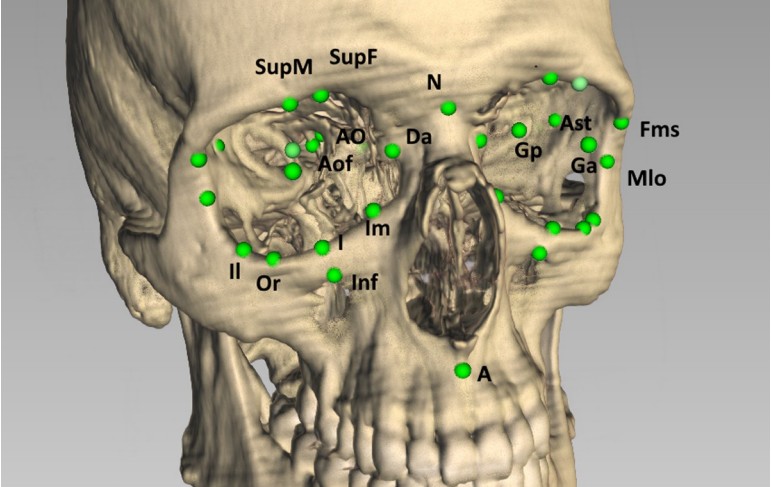

**Fig 1. Visualization of 3-D landmarks on a 3-D reconstruction of the skull made in FAT.**

**Table 2. Distances measured in the study and their definitions.**

| Distances | | | | | |
|---|---|---|---|---|---|
| posterior interorbital width | distance AO bilateral | bony Hertel FMS (bilateral) | Fms-Corpara-Ga | GlobeparalCor1 (bilateral) | Ga-Corpara-Mlo |
| Interorbitwidth posterior | distance Aof bilateral | Apex orbitae-Cor plane (bilateral) | AO-Cor | Endokantion RL | endo-endo |
| anterior interorbital width | distance Da bilateral | infraorbital depth (bilateral) | Inf-Cor | Exokantion RL | exo-exo |
| InterFrontomalarsut Distance | distance Fms bilateral | orbitdepht Aof (bilateral) | Aof-mA | Globe position anterior (bilateral) | Ga-Corpara-mMlo |
| InterMidpoint latorb Distance | distance Mlo bilateral | orbitdepth anterior (bilateral) | Da-Corpara-mMlo | Globe lateral position (bilateral) | mG-MSP |
| Orbital width (bilateral) | Da-Mlo | infraorbit-cor plane Supraorb (bilat.) | Inf-Corpara-SupM | Globe-Supraorbital plane (bilat.) | Ga-Corpara-So |
| Orbital height (bilateral) | Or-SupF | Mlo-coronal plane Da (bilateral) | Mlo-Corpara-Da | Globe-Or-plane (bilateral) | Ga-Corpara-Or |
| Orbit height 2 (bilateral) | Inf-SupM | Inf-coronal plane Da (bilateral) | Inf-Corpara-Da | Globe Protrusion (bilateral) | Ga-mA |
| orbitlength med (bilateral) | Da-AO | Sphenoidtrigone depth (bilateral) | Ast-Pst | midglobe-coronal plane Da (bilateral) | mG-Corpara-Da |
| orbitlength lateral (bilateral) | Mlo-AO | lateral orbit-Sphenoid anterior (bilat.) | Mlo-Ast | anterior globe coronal plane Da (bilat.) | Ga-Corpara-Da |
| orbitlenght superior (bilateral) | Aof-SupM | lateral orbit-Sphenoid posterior (bilat.) | Mlo-Pst | anterior globe coronal plane Mlo (bilat.) | Ga-Corpara-Mlo |
| orbitlenght inferior (bilateral) | Aof-Inf | Exokant-coronal plane (bilateral) | exo-Cor | midglobe-coronal plane Mlo (bilateral) | mG-Corpara-Mlo |
| orbitlenght lateral (Aof) bilateral) | Aof-Mlo | Endokantion-coronal plane (bilateral) | endo-Cor | Lid aperture (bilateral) | exo-endo |
| Fms-Cor (bilateral) | Fms-Cor | A Point-Midapertureplane (bilateral) | A-Corpara-mA | Globe coronar plane-N (bilat.) | Ga-Corpara-N |
| Mlo-Cor (bilateral) | Mlo-Cor | Supraorbital soft tissue-Cor plane (bilat.) | So-Cor | Globe-coronar plane-exo (bilat.) | Ga-Corpara-exo |
| Mlo-Midsag plane (bilateral) | Mlo-MSP | Orbitale soft tissue-Cor plane (bilat.) | Or-Cor | Globe position horizontal (bilateral) | mG-MSP |
| Fms-Midsag plane (bilateral) | Fms-MSP | Globe distance | distance Ga bilateral | Globe position vertical (bilateral) | mG-Fhpara-mA |
| Orbit depth (bilateral) | mA-AO | Globe diameter (bilateral) | Ga-Gp | midlat. orbit-cor. plane Supraorbit (bilat.) | Mlo-Corpara-SupM |
| Orbitdepth3 (bilateral) | mG-Corpara-mApex | lid midpoint-Apex orbitae (bilateral) | mL-AO | Dakryon-coronal plane Supraorbit (bilat.) | Da-Corpara-SupM |
| Orbitale-Cor (bilateral) | Or-Cor | midpoint Globe Cor plane-S (bilateral) | mG-Corpara-S | infraorbit-coronal plane Supraorbit (bilat.) | Inf-Corpara-SupM |
| Dakryon-Cor (bilateral) | Da-Cor | Globe-coronar plane-Endok (bilateral) | Ga-Corpara-endo | mid-lateral orbit-coronal plane Da (bilat.) | Mlo-Corpara-Da |
| Supraorbital foramen-cor. plane (bilat.) | Supf-Cor | MidGlobe-Apex orbitae (bilat.) | mG-AO | mid-lateral orbit-coronal plane Ao (bilat.) | Mlo-Corpara-AO |
| Nasion soft tissue-coronal plane | n-Cor | Globe posterior Apex (bilateral) | Gp-AO | Dakryon-coronal plane Apex orbitae (bilat.) | Da-Corpara-AO |
| Nasion-coronal plane | N-Cor | bony Globe protrusion (bilateral) | mA-Ga | supraorbit-cor. plane Apex orbitae (bilat. | SupM-Corpara-AO |
| mid Orbit-Sella plane | mMlo-Corpara-S | bony Gobe post protrusion (bilateral) | mA-Gp | infraorbit-coronal plane AO (bilateral) | Inf-Corpara-AO |
| mid Orbit-Basion plane | mMlo-Corpara-Ba | bony Globe mid protrusion (bilateral) | mA-mG | midpoint Mlo-coronal plane S | mMlo-Corpara-S |
| bony Hertel (bilateral) | Mlo-CorparaGa | Globe position sagittal (bilateral) | Aof-Ga | midpoint Mlo-coronal plane Ba | mMlo-Corpara-Ba |
| bony Hertel midaperture (bilateral) | Mlo-Corpara-mA | Globe-coronal plane (bilat.) | Ga-Cor | Midaperture-Cor (bilateral) | mA-Cor |

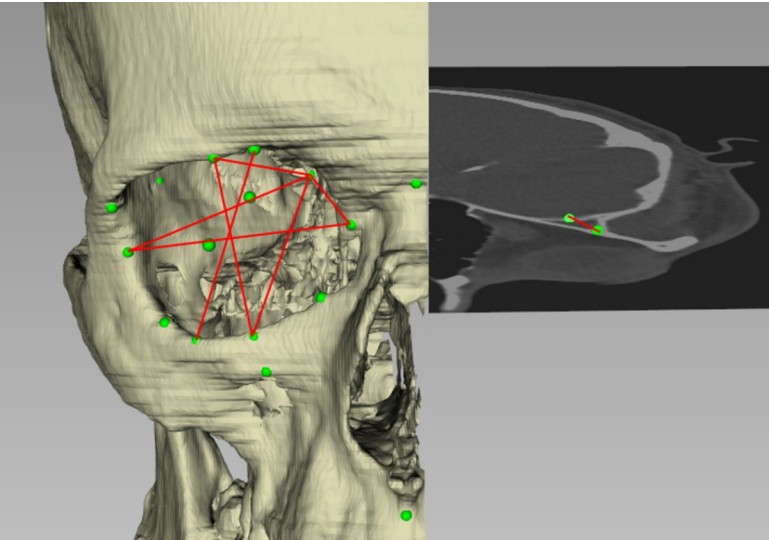

**Fig 2. Visualization of distances shown in Table 2.**

relation to the lateral orbital wall and *Dakryon*), and a more anterior placed orbit in relation to the middle cranial fossa. Regarding soft tissue differences in the non-EO group, men presented more protrusion of the supraorbital rim and *Exo-Endokanthion* in relation to the coronal plane. While these differences ranged from 1–2 mm for most parameters, supraorbital rim protrusion was marked with an average difference of 13 mm. Regarding angle measurements, only *medorbangl*e was different, being larger in women.

**Table 3. Comparison of EO versus nonEO CT scans.**

|  | EO | nEO |  | EO male | nEO male |  | EO female | nEO female |  |
|---|---|---|---|---|---|---|---|---|---|
|  | Mean ± SD | Mean ± SD | p | Mean ± SD | Mean ± SD | p(t) | Mean ± SD | Mean ± SD | p |
| **Volume/surface** |  |  |  |  |  |  |  |  |  |
| orbit volume | 29.9 ± 3.2 | 29.4 ± 2.7 | n.s. | 30.9 ± 3.4 | 30.4 ± 2.2 | n.s. | 29.3 ±3 | 27.5 ± 2.4 | 0.017 |
| orbit surface | 56 ± 3.8 | 55 ± 3.3 | n.s. | 57.3 ± 3.9 | 56.5 ± 2.7 | n.s. | 55.3 ± 3.7 | 52.7 ± 2.9 | 0.005 |
| **Distances** |  |  |  |  |  |  |  |  |  |
| orbit width | 42.1 ± 1.9 | 41.5 ± 2 | n.s. | 42.5 ± 1.5 | 42 ± 1.7 | n.s. | 41.8 ± 2.1 | 40.4 ± 2.3 | 0.020 |
| orbit height | 42.7 ± 2.2 | 41.4 ± 2.5 | 0.002 | 42.5 ± 1.9 | 41.5 ± 2.6 | n.s. | 42.8 ± 2.4 | 41.2 ± 2.3 | 0.009 |
| length lateral orbit (AO) | 47.8 ± 2.1 | 47.2 ± 2.4 | n.s. | 48.6 ± 2.3 | 47.8 ± 2.4 | n.s. | 47.4 ± 1.9 | 45.9 ± 1.9 | 0.004 |
| orbit depth (mA-AO) | 44.8 ± 2.4 | 44.3 ± 2.6 | n.s. | 45.6 ± 2.4 | 45.1 ± 2.6 | n.s. | 44.4 ± 2.3 | 43 ± 2.1 | 0.016 |
| Sphenoid trigone (Ast-Pst) | 10.3 ± 2.6 | 8.8 ± 2.8 | 0.003 | 10.2 ± 2.5 | 8.9 ± 2.9 | n.s. | 10.4 ± 2.7 | 8.8 ± 2.6 | 0.021 |
| Mlo-Ast | 17.9 ± 2.4 | 19.6 ± 2.5 | < 0.001 | 18.2 ± 2.3 | 20.2 ± 2.5 | 0.006 | 17.8 ± 2.5 | 18.5 ± 2.1 | n.s. |
| Mlo-Pst | 27.6 ± 3.2 | 27.5 ± 3.5 | n.s. | 27.7 ± 2.8 | 27.9 ± 3.7 | n.s. | 27.5 ± 3.4 | 26.6 ± 3.1 | n.s. |
| Mlo-cor1Aof | 34.5 ± 2.4 | 34.5 ± 2.6 | n.s. | 35 ±2.4 | 35.2 ±2.6 | n.s. | 34.3 ± 2.3 | 33 ± 2 | 0.040 |
| **Angles** |  |  |  |  |  |  |  |  |  |
| medorb angle | 19.8 ± 2.6 | 20.3 ± 2.8 | n.s. | 20.6 ± 2.1 | 19.7 ± 2.6 | n.s. | 19.4 ± 2.7 | 21.3 ± 3 | 0.009 |
| horizontal orbit angle | 51.5 ± 3.9 | 50.2 ± 2.7 | 0.043 | 51.3 ± 4.2 | 50.6 ± 2.5 | n.s. | 51.5 ± 3.8 | 49.6 ± 3 | 0.036 |
| **orbit-skull base** |  |  |  |  |  |  |  |  |  |
| posterior interorbital width | 30.3 ± 3.2 | 31.5 ± 3.4 | 0.049 | 31.6 ± 3.5 | 31.9 ± 3.1 | n.s. | 29.7 ± 2.9 | 30.9 ± 3.9 | 0.18 |
| anterior interorbital width | 20.6 ± 2.7 | 21.3 ± 2.7 | n.s. | 21.9 ± 2.3 | 21.1 ± 2.7 | n.s. | 19.9 ± 2.6 | 21.6 ± 2.6 | 0.016 |
| or-cor1 | 90.3 ± 4.3 | 88.6 ± 5.9 | n.s. | 91.7 ± 4.1 | 90 ± 5.7 | n.s. | 89.6 ± 4.2 | 85.9 ± 5.3 | 0.004 |

**Table 4. Sex differences.** Bony measurements include all CT scans, whereas soft tissue variables were made for the nonEO group only.

| | Mean±SD male | Mean±SD female | p | | Mean±SD Male | Mean±SD female | p |
|---|---|---|---|---|---|---|---|
| **volume/surface** | | | | **skull base middle cranial fossa—orbit** | | | |
| orbit volume | 30.6 ± 2.6 | 28.6 ± 2.9 | < 0.001 | mMlo-CorparaS | 47.8 ± 3.1 | 44.9 ± 2.5 | < 0.001 |
| orbit surface | 56.7 ± 3 | 54.3 ± 3.6 | < 0.001 | mMlo-CorparaBa | 68.9 ± 4.6 | 65.2 ± 3.4 | < 0.001 |
| **Distances** | | | | posterior interorbital width | 31.8 ± 3.1 | 30.2 ± 3.4 | 0.007 |
| orbit width | 42.2 ± 1.6 | 41.2 ± 2.2 | 0.012 | AO-Cor | 38.2 ± 3.7 | 36.8 ± 3.6 | 0.048 |
| inter-midlatorb (Mlo-Mlo) | 98.8±4.1 | 96.9±3.5 | 0.044 | SupM-Cor | 87 ± 4.9 | 82.9 ± 4.5 | < 0.001 |
| orbit length med | 44.7 ± 3 | 43.4 ± 2.7 | 0.014 | Or-Cor | 77.1 ± 4.3 | 74.2 ± 4.1 | < 0.001 |
| orbit length lat | 48.1 ± 2.3 | 46.8 ± 2 | 0.003 | Fms-Cor | 72.2 ± 4 | 69.4 ± 3.6 | < 0.001 |
| orbit depth | 45.2 ± 2.6 | 43.8 ± 2.3 | 0.003 | Mlo-Cor | 70.2 ± 4 | 67.6 ± 3.4 | < 0.001 |
| orbit length sup | 54.9 ± 2.7 | 53.1 ± 2.8 | < 0.001 | Fms-MSP | 50.4 ± 2.3 | 49 ± 1.6 | < 0.001 |
| orbit length inf | 50 ± 2.6 | 48.3 ± 2.8 | 0.001 | mA-Cor | 76.3 ± 4.1 | 73.7 ± 3.8 | < 0.001 |
| orbit lenght lat (AoF) | 48.1 ± 2.3 | 46.6 ± 2.3 | 0.001 | Da-Cor | 82.5 ± 4.7 | 79.9 ± 4.5 | 0.002 |
| orbit depth | 42.7 ± 2.4 | 41.3 ± 2.5 | 0.001 | or-Cor | 90.5 ± 5.4 | 88.1 ± 5 | 0.011 |
| Mlo-Ast | 19.6 ± 2.6 | 18 ± 2.4 | < 0.001 | **Soft tissue anatomy; only non-EO** | | | |
| Mlo-CorparaSupM | 16.9 ± 2.9 | 15.5 ± 2.5 | 0.011 | So-Cor | 95.8 ± 5.4 | 90.4 ± 5.8 | < 0.001 |
| Da-CorparaSupM | 47.4 ± 2.8 | 33.9 ± 2.1 | 0.003 | exo-Cor | 77.1 ± 4.7 | 74.3 ± 4.2 | 0.016 |
| Mlo-CorparaAof | 35.2 ± 2.5 | 33.7 ± 2.3 | 0.001 | endo-Cor | 85.4 ± 5 | 82.3 ± 4.8 | 0.013 |
| Da-CorparaAof | 44.3 ± 3 | 43 ± 2.7 | 0.014 | **Bulbus position** | | | |
| **Angles** | | | | midpointGlobe-CorparaS | 41.5±3.2 | 37.8±3.5 | < 0.001 |
| medorbangle° | 19.7±2.6 | 21.3±3 | 0.02 | Orbitdepth3 | 36.9±2.8 | 34.3±2.5 | < 0.001 |
| **Proptosis** | | | | Midglobe-AO | 50.5±2.8 | 47.8±2.8 | < 0.001 |
| Ga-So | 32±2.9 | 30±3.1 | 0.009 | mG-MSP | 54.5±3 | 51.9±2.6 | < 0.001 |
| Lidmidpoint (mL)-AO | 54.8±3.2 | 52±2.5 | < 0.001 | Ga-Cor | 86.4±5 | 82.6±5 | 0.003 |
| Ga-mA | 7.5±4.2 | 9.5±4.2 | 0.007 | | | | |
| Ga-corpara-N | 12.2±4.6 | 8.7±4.6 | < 0.001 | | | | |

Analysis concerning globe position showed a more anterior and lateral globe with respect to the coronal plane and *orbital apex* in men (i.e. a more anterior positioned globe in absolute measures to the coronal plane defined by *Porion* landmarks). Regarding clinically measured exophthalmos (i.e. the relative globe position to the orbital aperture), women presented a more anteriorly positioned globe regarding to *orbit midaperture* (2 mm) and *Nasion* (3.5 mm).

After investigating symmetry, EO characteristics, and sex differences, the last issue was to evaluate the influence of anatomy on orbital proptosis. As exophthalmos is a major stigma in EO, analysis of the non-EO group was of particular interest. Nevertheless, EO patients were evaluated, too, as anatomical factors could play a role in the extent of exophthalmos and thus in the decision on therapeutical measures.

The results are given in Tables 5 & 6. First, the correlation between exophthalmos and bony distances including orbit volume and surface was evaluated. The distance globe-*apex orbitae* correlated with orbital dimensions. This implies that in larger orbits the distance from anterior globe to apex is larger, too.

Similar results were found for EO and non-EO CT scans regarding the horizontal globe position. A more lateral position was correlated with a larger orbit and wider and higher orbital aperture, larger orbital depth, and a retropositioned lateral orbital rim. Concerning Hertel measures (i.e. relative exophthalmos in relation to the orbital aperture and *Nasion*), more correlations were found in the EO group. Generally, a larger orbit was associated with a more anterior globe position. Regarding the position of the orbital rim, larger Hertel values

**Table 5. Correlation of bony anatomic parameters (distances) to globe position values.**

| | Hertel analogue measures (Globe to Fms/Mlo/CorparaMlo/mMlo) | | | | anterior Globe to AOF/mApex | | | | anterior Globe to MSP | | | |
|---|---|---|---|---|---|---|---|---|---|---|---|---|
| | EO | | nEO | | EO | | nEO | | EO | | nEO | |
| | r | P | r | p | r | p | r | p | r | p | r | p |
| **General orbit size** | | | | | | | | | | | | |
| Volume | 0.31 | 0.031 | | | | | 0.41 | < 0.001 | 0.41 | 0.003 | 0.47 | < 0.001 |
| Surface | 0.32 | 0.022 | 0.27 | 0.029 | | | 0.49 | < 0.001 | 0.42 | 0.002 | 0.51 | < 0.001 |
| orbit width | 0.41 | < 0.005 | 0.45 | < 0.001 | | | 0.34 | 0.004 | 0.51 | < 0.001 | 0.53 | < 0.001 |
| orbit height | 0.36 | 0.008 | | | 0.27 | 0.050 | | | 0.29 | 0.040 | 0.25 | 0.033 |
| orbit length med | 0.38 | 0.006 | 0.37 | 0.002 | | | 0.45 | < 0.001 | | | | |
| orbit length lat | 0.28 | 0.045 | | | 0.44 | 0.001 | 0.51 | < 0.001 | | | | |
| orbit depth | 0.40 | 0.003 | 0.26 | 0.027 | 0.37 | 0.007 | 0.50 | < 0.001 | | | | |
| orbitlength sup | 0.31 | 0.026 | 0.37 | 0.001 | 0.61 | < 0.001 | 0.65 | < 0.001 | | | | |
| orbit length inf | 0.46 | < 0.001 | | | 0.54 | < 0.001 | 0.64 | < 0.001 | | | | |
| orbit depth Aof | 0.28 | 0.047 | | | 0.55 | < 0.001 | 0.71 | < 0.001 | | | | |
| infraorb depth | 0.42 | 0.002 | 0.34 | 0.004 | | | 0.46 | < 0.001 | 0.28 | 0.046 | 0.33 | 0.004 |
| **Position of orbital rim** | | | | | | | | | | | | |
| infmid-corSupra | 0.35 | 0.011 | 0.29 | 0.016 | | | | | | | | |
| latorb-corDa | 0.48 | < 0.001 | 0.64 | < 0.001 | | | | | 0.35 | 0.010 | 0.26 | 0.026 |
| latorb-AST | | | | | 0.35 | 0.012 | 0.27 | 0.022 | | | | |
| latorb-Pst | | | -0.33 | 0.005 | | | 0.28 | 0.020 | | | | |
| latorb-corSupra | 0.48 | < 0.001 | 0.69 | < 0.001 | | | | | 0.33 | 0.016 | 0.29 | 0.013 |
| Da-corSupra | 0.31 | 0.027 | | | 0.32 | 0.019 | | | | | | |
| inforbmid-corSupra | 0.35 | 0.011 | 0.28 | 0.017 | | | | | | | | |
| latorb-corAO | | | | | 0.33 | 0.016 | 0.46 | < 0.001 | | | | |
| Da-corAO | 0.40 | 0.003 | 0.39 | < 0.001 | | | 0.44 | < 0.001 | | | | |
| orbit depth ant | 0.40 | 0.003 | 0.57 | < 0.001 | | | | | 0.35 | 0.012 | | |

were found in a more retral positioned lateral rim (compared to the orbital aperture midpoint, the medial and superior borders) and a more retral inferior rim (compared to the superior rim).

Concerning orbital angles, most correlations were found for angles between lateral orbital wall and *midsagittal plane*. A larger angle between the lateral wall and the midsagittal plane was correlated to less proptosis, a more retral lateral orbital rim, and a more lateral globe in both groups. A larger *infraorbital slope angle* (i.e. a more caudal inferior rim) was associated with less proptosis and a more retral orbital rim position in the non-EO group. *Medorb angle* and *latorb angle* were associated with changes inherent to their definitions (e.g. a smaller *medorb angle* implies a smaller distance between left and right *Dakryon* and thus a smaller anterior interorbital width).

## Discussion

In the discussion of our findings, three aspects had to be considered: First, our method utilizing 3-D cephalometry, then the results regarding the comparison between patients with and without EO and lastly, sexual dimorphism and the influence of anatomy on globe position.

As 3-D cephalometry has already been proven to be reliable regarding intra- and interobserver reliability [16,17], our study did not include such measurements. 3-D cephalometry will yield an inter- and intraobserver landmark selection accuracy of less than 0.5 to 1mm in most

**Table 6. Correlations of angle measurements to globe position.** Hertel values were defined as: *Ga-CorparaMlo* and *Fms-CorparaGa*.

| | EO | nEO | | EO | | nEO | | EO | nEO | | EO | | nEO | | EO | | nEO | |
| --- | --- | --- | --- | --- | --- | --- | --- | --- | --- | --- | --- | --- | --- | --- | --- | --- | --- | --- |
| | r | r | P | r | p | r | P | r | r | p | r | p | r | p | r | p | r | p |
| **Proptosis bone** | latorb° | | | medorb° | | | | horizontal orbital° | | | infraorbslope° | | | | latorb-MSP° | | | |
| distance Mlo-Mlo | -0.26 | | 0.028 | 0.43 | 0.001 | 0.35 | 0.003 | | | | | | -0.31 | 0.009 | 0.47 | < 0.001 | 0.39 | < 0.001 |
| mG-CorparaS | | | | | | | | | -0.25 | 0.039 | | | -0.32 | 0.007 | -0.40 | 0.004 | -0.37 | 0.002 |
| Hertel-values | | | | | | | | | | | | | -0.25 | 0.036 | | | 0.38 | 0.001 |
| Ga-CorparamidApex | | | | | | | | | -0.24 | 0.044 | | | -0.28 | 0.017 | -0.30 | 0.030 | -0.29 | 0.014 |
| mG-MSP | | | | 0.51 | < 0.001 | 0.30 | 0.011 | | | | | | | | 0.42 | 0.002 | 0.39 | < 0.001 |
| Fms-Aperture | | | | 0.29 | 0.037 | | | | | | | | | | 0.39 | 0.005 | 0.48 | < 0.001 |
| **Orbit-skull base position** | | | | | | | | | | | | | | | | | | |
| mMlo-CorparaBa | | | | | | | | | -0.25 | 0.036 | | | -0.31 | 0.009 | -0.58 | < 0.001 | -0.45 | < 0.001 |
| post interorbit width | -0.25 | | 0.037 | | | | | | 0.32 | 0.007 | | | | | | | | |
| ant interorbit width | | | | 0.91 | < 0.001 | 0.90 | < 0.001 | | | | 0.36 | 0.009 | | | 0.34 | 0.014 | 0.33 | 0.005 |
| AO-Cor | | | | | | | | -0.24 | | 0.040 | | | | | | | | |
| Supra-Cor | | | | | | | | | | | | | -0.29 | 0.014 | -0.30 | 0.034 | | |
| Or-Cor | | | | | | | | -0.29 | | 0.016 | | | | | -0.31 | 0.026 | | |
| Mlo-Cor | | | | | | | | | | | | | | | -0.42 | 0.002 | -0.31 | 0.009 |
| Fms-MSP | -0.25 | | 0.038 | 0.51 | < 0.001 | 0.25 | 0.035 | | | | | | -0.28 | 0.019 | 0.58 | < 0.001 | 0.48 | < 0.001 |
| mA-Cor | | | | | | | | | | | | | -0.25 | 0.034 | | | | |
| Da-Cor | | | | | | | | | | | | | -0.28 | 0.020 | | | | |
| OrST-Cor | | | | | | | | | | | | | -0.30 | 0.012 | | | | |
| **Proptosis soft tissue** | | | | | | | | | | | | | | | | | | |
| Ga-CorparaExo | | | | | | | | | | | | | | | | | 0.28 | 0.017 |
| Lidmid-AO | | | | | | | | | | | | | -0.28 | 0.020 | | | | |

landmarks [18–21] and up to 1.9 mm in less identifiable landmarks [22], which could be described as Bookstein type 3 [20,23,24]. We addressed this issue by basing our cephalometric analysis on well-replicable landmark points belonging to Bookstein types 1 and 2 in most instances [23,24]. Adding 3-D surface reconstructions for landmark placement (i.e. a combination of 2D and 3D surface views as implemented in our software) will further add reliability [21,25]. As a means of metric analysis, 3-D cephalometry as performed in this investigation can thus be judged as being highly reliable regarding landmark placement as well as linear and angular measurements [26] and well suited for orbital and periorbital analysis.

Further limitations in accuracy lie in the CT slice thickness, which we addressed by excluding CTs with a slice thickness larger than 1.5 mm and the positioning of landmarks. Although the latter is relatively reliable (as mentioned above), it is not without inaccuracy as Smektała et al. [22] found in their systematic review of the experimental and clinical assessement of three-dimensional cephalometry. Thus, the use of cross-hairs and the possibility to combine 2-D CT images with 3-D reconstructions for landmark placement as implemented in our investigation should improve the accuracy of our investigation. For further control, several well documented variables were included that did not primarily serve for our analysis (e.g. *SNA angle* and *BaSN angle)*. All variables lay within given normative ranges (*SNA angle*, *BaSN angle*; Table 7 [27–29]. The same applied to well-documented study parameters such as orbital volume in non-pathologic groups.

Regarding symmetry, the differences in this investigation were clinically insignificant and lay within the accuracy range of landmark placement. This observation is in line with previous evaluations that have led to the use of virtual mirroring in the preoperative planning of orbital

**Table 7. Results for distances in derived from previous studies.** Mean Orbit Volume is given in ml, Mean Orbit Surface in cm$^2$, all further parameters are in mm. Studies on non-Caucasian groups are marked "*".

| | All Groups | Male | Female |
|---|---|---|---|
| | Literature | Literature | Literature |
| Orbit Volume (cm$^3$) | 24.3[43]* 25.6[62]* 26.8[63] 25–28.9[13,45–47] 27.7[64] 33.2[65] | 26.8[49] 27.8[66] 29.6[50] 29.2[64] | 23.2[51] 26.5[49] 25.6[66] 25.9[64] |
| Orbit width | 39.7[67]*39.8[52]– 41.3[53] 41.7[64] | 35[68] 39.8[54] 41[69] 42.5[64]44.2[55] | 33.6[68] 36.9[54] 39[69] 41[64] 42.1[54] |
| Orbit height 2 (Inf-SupM) | 33.4[52] 34.1[64] 36.2[67]*36.6[45] | 32.4[10] 33.6[64] 35.9[56] 36.6[68] 39[69] | 31.8[10] 34.1[64] 35.4[56] 35.7[68] 37[69] |
| Orbit lenght medial | 41.3[61]*[64]43[57] 46.9[58] | 41.9[44]* 42.3[64]43.6[57] 48.6[55] | 40[64] 40.7[44]* 42.4[57] 46.2[55] |
| Orbit lenght lateral | 43.8[57] 47.2[53] 47.2[64] | 42.9[70] 44.3[57] 48.3[64] 50[55] | 43.2[70] 43.3[57] 45.7[64] 48.3[55] |
| Orbit depth (mA-AO) | 42[59] 48.8 [64] | 43[53] 45.2[60] 51[64] | 40.5[51,53] 42.8[60] 47[64] |
| Orbit length superior | 47.3[64] 50.5[53] | 48.6[64] 51.8[16] 53.9[55] | 45.6[64] 51.5[55] 51.7[16] |
| Orbit length inferior | 48.1[64] 49.6[61] | 47[16] 49.3 [64] 50.5[61] | 46.6[64] 46.9[16]– 48.7[61] |
| Ant. sphenoid depth (Mlo-Ast) | 17.9[43]*20.4[67]* | 26.3[41]* | 24.8[41]* |
| MLO-Pst | 25.5[41]* | | |
| ant. interorbital with | | 25.2[68] | 24[68] |
| post. Interorbital width | 26.7[67]*# | | |

and midface reconstructive surgery in fractures of the orbit and midface and in navigated surgery [12,30–33]. In all investigations, most side differences were rated less than 1 mm with maximum differences less than 2 mm. It has to be stated that this applies to measurements on bone. However, facial soft tissue measurements and symmetry analysis has shown that bilateral tissue thickness differences lie between 0.3–0.5 mm thus symmetry findings should apply to soft tissue measurements as well [34]. Even if the use of the unaffected side's anatomy as a model for the reconstruction of the affected side is not without limitations [35], it is still the most widely used procedure which is supported by our findings.

While non-EO CT scans are highly symmetrical, the situation in EO is different. Globe asymmetry (i.e.more than 2 mm difference [36], has been shown to indicate a more severe and active disease [37]. Thus, 3D-cephalometry and the respective analysis would be the ideal means to exactly analyze the extent and progress of EO to decide on appropriate therapeutic means.

Concerning anatomical differences between EO and non-EO patients, Baujat et al. [8] found a slightly smaller lateral orbital angle (corresponding with the angle between lateral orbital wall and midsagittal plane in this investigation) and a smaller interorbital distance in patients with EO compared to non-EO patients with exorbitism. Although they did not compare EO to non-exophthalmos non-EO CT scans and did not evaluate sex influences, the authors concluded that while minor anatomical differences might exist, orbital anatomy is, at most, of little relevance to the etiology of EO [8].

In our study, 15 parameters including 2 angles showed statistically significant differences between patients with and without EO in women and only one in men. Why men presented less anatomical differences remains speculative. A possible explanation is that the increase of soft tissue within the orbit might lead to bony resorption and remodeling leading to larger dimensions. The combination of smaller orbit size and potential differences in sex-specific bone structure [38] could thus lead to more remodeling in thinner bone in women (i.e. especially the medial wall) which would explain the larger orbit volume and smaller interorbit distances (Table 3) in women with EO. Thus, a possible explanation of the found anatomical differences in women is that these are not a cause of EO but rather a consequence. This

corresponds to the mechanism of auto-decompression in EO concerning the medial and inferior orbital walls [39,40]. The lateral orbital angle *(latorbline–midsagittal plane)* which Baujat et al. [8] described as higher in patients with EO did not differ in our study. Instead, it was significantly larger in women. As no analysis of the influence of sex was included in the research of Baujat et al. [8], we suppose that the difference concerning this angle was rather a display of sexual dimorphism than a parameter relevant to the etiology of EO. We agree with Baujat's et al. [8] finding that the orbital anatomy is generally very similar in patients with and without EO even though there might be minor differences in orbital geometry. Thus, orbital anatomy is unlikely to have a meaningful role in the etiology or extent of EO.

On the other hand, anatomical data is important regarding surgical interventions as pointed out by Lee et al. [41] and Takahashi et al. concerning the deep lateral and medial bony orbital regions [42]. Thus, the distance from the lateral orbital rim and the increased dimensions of the sphenoid trigone as found in this study can be relevant in surgical EO treatment. This was stressed by Shin et al. [43] who investigated safe zones for decompression surgery in the lateral orbit in non-EO CT-scans, although their data on the sphenoid trigone cannot be compared with our findings due to different measurements. Furthermore, an analysis of anatomical parameters affecting the outcome of decompression surgery in EO showed that sphenoid trigone removal was an independent important factor [44]. Comparing EO to non-EO data, a statistically significant difference regarding the sphenoid trigone was seen in our study highlighting the importance of this structure.

Our data concerning orbital anatomy is generally in line with the results of previous evaluations of orbital anatomy in non-EO patients. Concerning distances and angles, only studies with similar definitions of the particular parameters were included in our comparison tables. Matching with data of EO investigations was difficult, as no data for most of our variables exists to our knowledge. Regarding non-EO data, all comparable measurements matched our findings with little differences concerning distances [13,16,45–70] or angles [8,27–29,45,49,71–73] in studies performed on patients with Caucasian ethnicity. Previously published findings on orbital anatomy are presented in Tables 7 & 8 for comparison with our results. As many studies on orbital anatomy have been performed on patients with different ethnic background, mostly from Asian countries, these have been marked in Table 7. A detailed discussion on ethnic differences in orbital anatomy (e.g. smaller and shorter orbit) will not be discussed as this would be off topic and overstretch the scope of this study.

Regarding sexual dimorphism in orbital anatomy, previous studies have shown statistically significant differences in orbital volume with men having larger orbit volumes [16,47,48,64] as well as in orbital morphology [11,16,56,60,64]. Regarding the shape of the orbital aperture–the coefficient derived from orbital width and height—which has been described as more round in women [64], our study showed no statistically significant sex-based difference even if the orbital aperture showed a slightly larger height/width ratio in women.

**Table 8. Results for angles for comparison from previous studies.**

|  | All Groups | Male | Female | EO | Non-EO |
|---|---|---|---|---|---|
|  | Literature | Literature | Literature | Literature | Literature |
| **SNA (°)** |  | 80.8[27] 82.5[27] | 80.5[28]– 82.8[27] |  |  |
| **BaSN (°)** | 127.1[29] 133.5[29] |  |  |  |  |
| **Latorbangle (°)** | 90[8] |  |  |  |  |
| **Orbit angle horizontal (°)** | 45[71] 53.6[45] | 47.9[72] 53.6[49] | 48.1[72] 50.8[49] |  |  |
| **Latorbline–Midsagittal plane (°)** | 45[73] |  |  | 42[8] | 40[8] |

In our study (Table 4), 41 variables were significantly different, most being larger in men with an average difference of 1–2 mm, leading to a 2 ml larger orbit volume. A larger bony orbit in men has been confirmed in previous studies, too, with an identical difference of ca. 2 ml [16,47,63]. Only orbital proptosis measured in relation to the orbital aperture was larger in women. Additionally, statistically significant differences existed in 1 angle parameter. Regarding orbital volume measurements, several methods have been suggested [13,74,75] which will lead to somewhat different results as the anterior confinement is calculated by way of different approaches (which in many reports are proprietary algorithms of the respective authors and not publicly available). Furthermore, volume calculation depends on the CT reconstruction strategy where computing on axial scans leads to larger results [62]. Thus, the results yielded by our approach lie in the upper range of reported orbital volume sizes ranging from 24.3–33.2 ml (Table 7). Regarding orbit size, smaller values were presented in studies on patients of Asian ethnicity (Table 7). Deveci et al. found an average of 28.4 ml [46], Regensburg et al. 28.9 ml in Caucasian men [48], whereas the findings of Graillon et al. and Wagner et al. lay at over 26 ml [13,47] and Adenis et al. found an average of 25.6 ml in women and 27.8ml in men [66]. Besides different volume size, however, the gender difference of 2 ml was similar [16,47,66] to our results.

Whether the differences between male and female orbits would be mostly due to scaling or more in shape was not addressed in this study, but will be an issue of further investigations. Regarding periorbital anatomy, men presented a statistically significant far more pronounced supraorbital rim. This is in line with previous investigations on facial sexual dimorphism [76,77], and is an important topic in transgender facial feminizing surgery [78] or in esthetic facial surgery in men [79]. The influence on the orbital rim on perceived exophthalmos will be discussed below.

The last issue concerns the relation of globe position to the anatomy of the orbit, especially the orbital aperture. Regarding influential factors on exophthalmos, several studies have shown that globe protrusion diminishes with age while the orbital aperture size increases [80–83]. Furthermore, Body Mass Index (BMI) seems to play a role [84] as increased BMI is reported to correlate with globe protrusion. Interestingly, no study could be found by the authors that tried to correlate bony anatomic measures with globe position as performed in this study. Only one investigation by Shin et al. [85] measured distance relations of the orbital rim to the globe in a Korean study group, but did not provide further statistical analysis, whereas Kim et al. [86] reported on the significance of orbital rim to globe relation for forensic reconstructions.

Thus, our report is the first known to the authors that correlates orbital rim configuration and orbit measures to globe position (Fig 3). Our findings regarding globe position and anatomy can be differentiated in two aspects. First, internal bony measurements and second, importance of the bony aperture configuration. Looking at intraorbital measurements, the finding that longer orbits correlate with longer distances between orbital apex and globe may seem trivial but they are not. By now, no correlation of bony size and size of the soft tissue contents is known to the authors. The only known investigation to the authors is the study of Li et al. [72], who investigated the correlation of orbital volume and orbital wall length to ocular protrusion. Similarly to our investigation, a positive correlation of orbital volume and a more forward globe position was identified. However, the authors did not specify how exophthalmos was measured. From the given values it can be inferred that they used a calculated Hertel value and not the true distance of the globe to the orbital apex as performed in our investigation.

The opposite result, i.e. that a larger orbit might favour a more retropositioned globe or a "shallow" orbit would statistically be correlated to more exophthalmos would also have been understandable. It is a truly novel finding of this investigation that this was not the case. Secondly, our results demonstrate the importance of the orbital rim configuration on visually

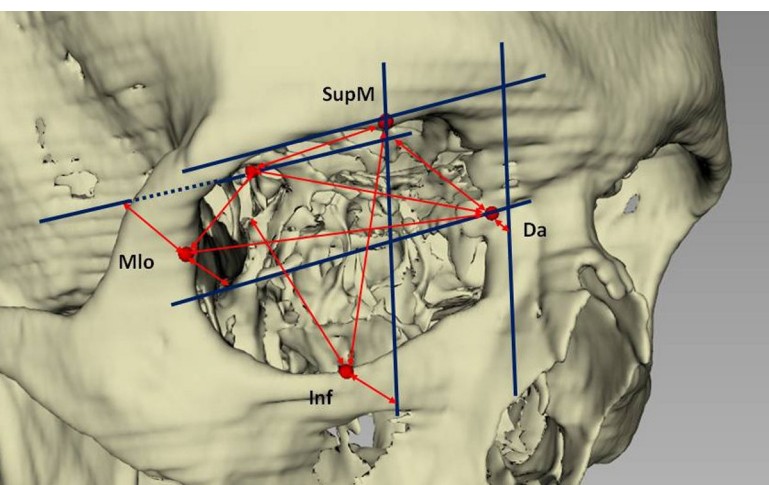

**Fig 3. Visualization of anatomic parameters defining globe position measurements.** An increase of the displayed distances (as stated in red arrows) anterior to the orbital aperture will lead to increased Hertel values, whereas an increase of distances within the orbit will lead to larger distances of the globe to the orbital apex. Analysis planes (parallel planes to *Cor through Da*, *SupM,* and *AO* are visualized as blue lines).

perceived exophthalmos, even if the "true" position of the globe to the orbital apex is unchanged. In a clinical setting, both Hertel and Naugle exophthalmometers cannot be used for this task as they rely on horizontal bony structures (lateral rim & *Nasion*) or supra- and infraorbital rim to define globe position. Furthermore, an EO-associated increase in periorbital soft tissue thickness has been described [87,88], which might interfere with the correct use of clinical exophthalmometers (i.e. a more anterior position of the device). Using an elaborate CT analysis, we could show the influence of all orbital rims. Even though it might seem self-explanatory that a retruded lateral orbital rim would increase Hertel values without the change of the globe, this has not been reported on before. Our results showed correlation coefficients in the non-EO group of 0.57–0.69 which can be interpreted as moderate to strong. Takada et al. have presented a hypothesis how a retruded lateral orbital rim and perceived exophthalmos could be interdependent as they suggested that a more forward positioned globe due to an enlarged ethmoid (an increase of anterior and posterior interorbital width as seen in EO patients in this study) could inhibit the growth of the lateral orbital wall [67].

From a clinical viewpoint, this is of importance as retruded orbital rims mimic exophthalmos due to other reasons and can be a cause for surgical intervention. Thus, in non-EO patients, polyethylene implants are used to augment a retruded orbital rim [89]. In EO treatment, the lateral rim may be advanced-rotated forward [15,90]. Utilizing our 3-D cephalometric analysis, it is possible to distinguish between real exophthalmos (i.e. increased distance globe to orbital apex) and pseudoexophthalmos were the globe remains unaltered but the orbital rim is deficient. This knowledge might be important in choosing the appropriate surgical therapy if an intervention is planned.

As EO is a relatively rare disease, the number of patients which can be included is limited. If it was possible to include more patients, the statistical analysis could be even more convincing. Still, our analysis included more patients than the majority of evaluations of orbital anatomy (Baujat et al., 2006: 105 patients; Weaver et al., 2010: 39 patients; Moon et al., 2020: 24 patients; Ji et al., 2010: 64 patients) [8,10,11,16] and should thus be more statistically robust.

As our research included both a larger number of patients as well as the largest number of evaluated anatomical parameters using 3-D cephalometry, our results should not be seen as redundant but as a confirmation of supposed anatomical conditions.

Out of 28 previous evaluations referenced (Tables 7 and 8), positioning of 3-D landmarks on 3-D reconstructions of the bony orbit was carried out in 10 cases according to the respective methods chapters. Due to the reproductibility of 3-D landmarks, we suppose that methods of orbital analysis utilizing 3-D landmarks on 3-D objects are superior to alternative approaches e.g. measuring directly in 2-D or on 3-D reconstructions regarding precision and comparability.

Thus, we hope to provide useful data in both surgical planning and further research on orbital anatomy and EO. As Borumandi et al. pointed out in a review paper, sound anatomical knowledge is a prerequisite for surgical success in EO decompression surgery [91], and this study furnishes new anatomical data. The sphenoid trigone and its anterior-posterior dimensions seem to be of special importance as surgical parameters in decompression surgery discussed in Cruz et al. 2021 [92] and described as an important surgical factor by Rajabi et al. [44]. Furthermore, the data provided on orbital rim anatomy, especially supraorbital rim protrusion and lateral rim retrusion can be important in planning EO decompression surgery [15].

All patients included in this research were of Caucasian ethnicity. As differences in orbital anatomy between ethnicities exist [11,52], it could be of interest to examine whether our observations are valid in other ethnicities as well.

## Conclusions

The relevance of orbital anatomy in EO was investigated utilizing 3-D cephalometry in a Caucasian population. As only few statistically significant differences between EO and non-EO patients were found, anatomy seems to be of minor importance at best. Regarding sex, our investigation confirmed known aspects and added new 3-D data. Gender-related anatomical differences regarding orbital size and the configuration of the orbital aperture (e.g. the more prominent supraorbital rim in men) may have an impact on the visual appearance of EO patients and trigger therapeutical demands. This investigation introduced the interdependence of relative (i.e.to the orbital aperture) and absolute (to the orbital apex) globe position on anatomical parameters. Finally, the anatomical data provided should be helpful in surgical decision making and treatment planning concerning EO decompression surgery.

## Ethics statement

Our study was approved by the Leipzig University ethics commission (No. 285-14-25082014) and performed in accordance with the Declaration of Helsinki.

## Supporting information

**S1 Data.**
(XLS)

## Author Contributions

**Conceptualization:** Ina Sterker.

**Data curation:** Konstantin Volker Hierl, Matthias Krause.

**Investigation:** Konstantin Volker Hierl.

**Software:** Daniel Kruber.

**Supervision:** Matthias Krause, Ina Sterker.

**Visualization:** Konstantin Volker Hierl.

**Writing – original draft:** Konstantin Volker Hierl.

**Writing – review & editing:** Ina Sterker.

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
