## [Decision Letter · Decision Letter 0]

10 Dec 2021

PONE-D-21-332223-D cephalometry of the the orbit regarding endocrine orbitopathy (EO), exophthalmos, and sexPLOS ONE

Dear Dr. Hierl,

Thank you for submitting your manuscript to PLOS ONE. After careful consideration, we feel that it has merit but does not fully meet PLOS ONE’s publication criteria as it currently stands. Therefore, we invite you to submit a revised version of the manuscript that addresses the points raised during the review process.

Dear Author,Please go through the feedback and upload the revised file as per feedbackThank you

We look forward to receiving your revised manuscript.

Kind regards,

Kapil Amgain

Academic Editor

PLOS ONE

Journal Requirements:

Reviewers' comments:

Reviewer's Responses to Questions

**Comments to the Author**

1. Is the manuscript technically sound, and do the data support the conclusions?

Reviewer #1: Partly

Reviewer #2: Yes

2. Has the statistical analysis been performed appropriately and rigorously? 

Reviewer #1: Yes

Reviewer #2: Yes

3. Have the authors made all data underlying the findings in their manuscript fully available?

Reviewer #1: Yes

Reviewer #2: Yes

4. Is the manuscript presented in an intelligible fashion and written in standard English?

Reviewer #1: Yes

Reviewer #2: Yes

5. Review Comments to the Author

Reviewer #1: The author’s work is appreciated but it would have been better if they incorporate these suggestions to make this manuscript even better.

1. Please remove the short form (EO) from the title as full form is already mentioned, if possible.

2. In abstract section, please do not start the sentence with number rather start with words like- ‘two’ instead of ‘2’. Please make the conclusion part concise and short.

3. Title in the beginning and in the main manuscript is different. Please calrify.

4. Please mention the objective of this study at the end of ‘introduction part’ after it’s rationale.

5. Out of 123 samples, 71 were having no known pathology, doesn’t it raise the ethical concern for radiation exposure while doing CT? Please defend this.

6. Please mention your own findings only in the result section.

7. In discussion, please try not to repeat the result findings again if it's not applicable for discussion with other literatures and also you can remove the reference table and figures from the discussion as it’s already there in the results.

8. There is inadequate discussion on the interpretation and implication of study findings. It would have been better if they have included more comparable studies on discussion.

9. Please correct some grammatical error in the manuscript.

10. Please write the conclusion part in short, clear and concise.

Reviewer #2: Portions of the results section have been placed in the methods section. The method section basically focus on the research design, how the study was conducted and how the reader who wants to do the similar study can reproduce the method in this paper

6. PLOS authors have the option to publish the peer review history of their article (what does this mean?). If published, this will include your full peer review and any attached files.

Reviewer #1: No

Reviewer #2: No

---

## [Author Response · Author response to Decision Letter 0]

5 Jan 2022

Dear Mr Amgain, dear reviewers,

Thank you very much for your comments to improve the quality of the submitted paper draft.

In the following chapters, all remarks will be covered and the text changes marked appropriately in the paper and in this document.

Remarks by the editor:

This has been done.

This study has been a retrospective investigation on medical records. The local University ethics commission waived the requirement of informed consent as pseudonymization of all data was performed. Thus the following sentence was included on p. 10 at the end of the Materials and methods section (highlighted in green):

This retrospective study was approved by the local Leipzig University ethics commission (No. 285-14-25082014) and followed the Declaration of Helsinki on medical protocols and ethics. As pseudonymization of the CT data was performed, the requirement of informed consent was waived by the ethics committee. 

Reviewer’s Comments to the Author

Reviewer #1: The author’s work is appreciated but it would have been better if they incorporate these suggestions to make this manuscript even better.

1. Please remove the short form (EO) from the title as full form is already mentioned, if possible.

The short form (EO) was removed from the title.

2. In abstract section, please do not start the sentence with number rather start with words like- ‘two’ instead of ‘2’. Please make the conclusion part concise and short.

2 was changed to as required:

…and without EO. Two out of 12 angles showed a statistically ….

The conclusion was shortened made more focused. The changes were (added: light blue, deleted words: red):

There appears to be In this study little difference in orbital anatomy between patients with and without EO was found…. Thus, orbital anatomy is likely to be of minor relevance in the etiology of EO but may influence therapy regarding timing and surgical procedures as orbital rim anatomy it affects exophthalmos. Sex-based differences were confirmed.

3. Title in the beginning and in the main manuscript is different. Please clarify.

As the title fulfills the word count requirements of the short title, the short title was changed to the full title.

4. Please mention the objective of this study at the end of ‘introduction part’ after it’s rationale.

The objectives of our investigation have now been summarized at the end of the introduction. The changes are:

… The purpose of our study was to evaluate orbital anatomy in patients with and without EO investigating the relevance of orbital anatomy in the etiology of EO and exophthalmos – one of its major clinical features – using a new approach of three-dimensional cephalometric measurement. In previous investigations Utilizing measurements in CT-scans, Baujat et al. and Rajabi et al. [8, 9] did not find such significant anatomical differences in EO patients besides slight differences in the lateral orbital angle (angle between the midsagittal plane and the lateral orbital wall) or interorbital distance. These studies, however, were limited by several factors like the use of axial CT slices for 2-D measurements instead of 3-D cephalometry [8, 9]. Baujat et al. compared two patient groups with exophthalmos (EO and non-EO) [8] whereas Rajabi et al. evaluated a small groups of non-EO patients [9]. The number of observed parameters was limited in both evaluations. Moreover, the influence of sex was not evaluated. 

Thus, the purpose of our study was to evaluate orbital anatomy in patients with and without EO investigating the relevance of orbital anatomy in the etiology of EO and exophthalmos – one of its major clinical features – using a new approach of three-dimensional cephalometric measurement.

5. Out of 123 samples, 71 were having no known pathology, doesn’t it raise the ethical concern for radiation exposure while doing CT? Please defend this.

The control CTs without anatomical discernable pathology had been acquired because of the following reasons: search for foci in the head & neck region due to general illnesses (but none found), search for anatomical causes of neurologic disorders (none found), CT scans in preparation of oral surgery/oral maxillofacial surgery which didn´t cause anatomical alterations with regard to this investigation (e.g. wisdom teeth removal)

The following sentence was added in Materials and methods section:

…group. The CT scans without anatomical discernable pathology (reference group) had been acquired in search for foci in the head & neck region due to general illnesses, in search for anatomical causes of neurologic disorders, and in preparation of oral surgery or oral and maxillofacial surgery not associated with anatomical alterations relevant to this investigation. Besides…

6. Please mention your own findings only in the result section.

The results regarding symmetry which had been repeated in the discussion were deleted.

The text was changed to:

…Generally, the orbital anatomy was highly symmetrical with an average side difference in distances of 0.3 mm and an average side difference in angles of 0.6° Regarding symmetry, the differences in this investigation were which is clinically…

7. In discussion, please try not to repeat the result findings again if it's not applicable for discussion with other literatures and also you can remove the reference table and figures from the discussion as it’s already there in the results.

Both tables in the discussion section have been changed to only display reference data. Repetitions of our results were removed from the discussion section.

8. There is inadequate discussion on the interpretation and implication of study findings. It would have been better if they have included more comparable studies on discussion.

Within the discussion section, some aspects have now been treated more in depth. 

This includes methodology: 

- 3D cepalometry per se (p. 16-17)

 and results:

- 3D symmetry (p. 18-19)

- 3D anatomy (p. 20-21)

- sex differences (p. 23-24)

- importance of orbital rim morphology (p. 24-26)

- importance for surgical treatment (p. 27)

More literature was included in the discussion (see references marked in blue)

9. Please correct some grammatical error in the manuscript.

Grammar errors have been deleted and style was improved.

10. Please write the conclusion part in short, clear and concise.

The conclusion was changed to be more concise and clear.

Reviewer #2: Portions of the results section have been placed in the methods section. The method section basically focus on the research design, how the study was conducted and how the reader who wants to do the similar study can reproduce the method in this paper

The materials and methods chapter was checked to include no results. Only data on the patient and control group was provided. Furthermore, only the lists of the utilized landmarks and the measured distances and angles have been presented. Concerning data presentation in the discussion (see reviewer 1/6 & 7), this has be changed.

---

## [Editor Report · Decision Letter 1]

18 Jan 2022

PONE-D-21-33222R13-D cephalometry of the the orbit regarding endocrine orbitopathy, exophthalmos, and sexPLOS ONE

Dear Dr. Hierl,

Thank you for submitting your manuscript to PLOS ONE. After careful consideration, we feel that it has merit but does not fully meet PLOS ONE’s publication criteria as it currently stands. Therefore, we invite you to submit a revised version of the manuscript that addresses the points raised during the review process.

We look forward to receiving your revised manuscript.

Kind regards,

Kapil Amgain

Academic Editor

PLOS ONE

Dear Author,

Please find the attached feedback file and revise the manuscript as per the comments.

---

## [Author Response · Author response to Decision Letter 1]

5 Feb 2022

The following text states the changes made according to the reviewer’s suggestions. Changes are highlighted in yellow. Changes requested by both reviewers were not independently highlighted.

Title page

3-D cephalometry of the the orbit in patients with and without endocrine orbitopathy

(EO). Which is the orginal title of study?

Two titles were initially given as a secondary short title was requested. As the original title did not exceed the word count of the short title, the short title was changed to the main title.

It is better to aviod abbreviations in title of study in such way?

The abbreviation was deleted

Abstract

What was the study design? Which statistical software and tools were used to analyze the data?

Information was added in the “methods” chapter of the abstract:

… Orbital anatomy of 123 Caucasian patients (52 with EO, 71 without EO) was examined using computed tomographic data and FAT software for 3-D cephalometry. Using 56 anatomical landmarks, 20 angles and 155 distances were measured. MEDAS software was used for performing connected and unconnected t-tests and Spearman´s rank correlation test to evaluate interrelations and differences. …

Main text

Grammar and typing errors were corrected

Materials and Methods 

The methodology was started with the study design as suggested.

The highlighted passages regarding patient data were transferred to the results section.

References

The capitalization of titles was changed from the respective journal style to PLOS reference style.

The title of reference 5 was not changed as capitalization of nouns is standard German grammar.

---

## [Editor Report · Decision Letter 2]

1 Mar 2022

3-D cephalometry of the the orbit regarding endocrine orbitopathy, exophthalmos, and sex

PONE-D-21-33222R2

Dear Dr. Hierl,

We’re pleased to inform you that your manuscript has been judged scientifically suitable for publication and will be formally accepted for publication once it meets all outstanding technical requirements.

Kind regards,

Dr. Kapil Amgain

Academic Editor

PLOS ONE

---

## [Editor Report · Acceptance letter]

4 Mar 2022

PONE-D-21-33222R2 

3-D cephalometry of the the orbit regarding endocrine orbitopathy, exophthalmos, and sex 

Dear Dr. Hierl:

I'm pleased to inform you that your manuscript has been deemed suitable for publication in PLOS ONE. Congratulations! Your manuscript is now with our production department. 

Kind regards, 

on behalf of

Dr. Kapil Amgain 

Academic Editor

PLOS ONE